
# Cavity ring-down spectroscopy of water vapor in the *near*-UV region

Qing-Ying Yang[1], Eamon K. Conway[2,3], Hui Liang[1], Iouli E. Gordon[2], Yan Tan[1], and Shui-Ming Hu[1]

[1]Hefei National Laboratory and Department of Chemical Physics, University of Science and Technology of China, Hefei, 230026 China
[2]Center for Astrophysics, Harvard and Smithsonian, Atomic and Molecular Physics Division, Cambridge, MA, 02138 USA
[3]Kostas Research Institute for Homeland Security, Burlington, MA, 01803 USA

**Correspondence:** Yan Tan (tanyan@ustc.edu.cn)

**Abstract.**

Water vapor absorption in the near-ultraviolet region is essential to describe the energy budget of Earth, but little spectroscopic information is available since it is a challenging spectral region for both experimental and theoretical studies. A continuous-wave cavity ring-down spectroscopic experiment was built to record absorption lines of water vapor around 415 nm. With a minimum detectable absorption coefficient of $4 \times 10^{-10}$ cm$^{-1}$, 40 rovibrational transitions of H$_2^{16}$O were observed in this work, and 27 of them were assigned to the (224), (205), (710), (304), (093), (125), and (531) vibrational bands. A comparison of line positions and intensities determined in this work to the most recent HITRAN database is presented. Water vapor absorption cross-sections near 415 nm were calculated based on our measurements, which vary between $1 \times 10^{-26}$ and $5 \times 10^{-26}$ cm$^2$/molecule. These data will also significantly impact the spectroscopy detection of trace gas species in the *near*-UV region.

## 1 Introduction

While water is a central key for terrestrial life, it is also the most abundant greenhouse gas on Earth and plays an immense role in climate evolution. Understanding the water vapor rovibrational spectrum is the cornerstone to constructing a reliable water transmittance model in solar radiation. However, the water vapor absorption in the near-ultraviolet (*near*-UV) region is still poorly understood, which has become the remaining problem of missing absorption in the model of the earth-atmosphere energy balance Learner et al. (1999); Callegari et al. (2002); Tennyson et al. (2013). Accurate spectral information of *near*-UV water absorption is also required in both ground-based observations and satellite missions when retrieving other atmospheric absorbers based on spectroscopy in the UV/VIS region Yin et al. (2021); Lampel et al. (2015, 2017); Chance (2005); Orphal and Chance (2003). For instance, the peak of NO$_2$ absorption cross-sections is located at 400-420 nm, and the remote sensing of ozone relies on its absorption/emission in the *near*-UV as well Chance (2005); Orphal and Chance (2003).

Recent theoretical calculations and experimental measurements have improved the quality of the spectroscopic data set of the water molecule in the visible and *near*-UV. Tremendous efforts have been made both theoretically and experimentally to construct the database of water transitions Rothman et al. (2010); Bernath et al. (2010); Tennyson et al. (2009, 2013, 2016); Polyansky et al. (2018). However, the water absorption in the *near*-UV region is extremely weak compared to that in the





infrared region Gordon et al. (2017), which makes it more difficult to measure or model. Several earlier studies were carried out using high-resolution Fourier-transform spectroscopy (FTS) Carleer et al. (1999); Coheur et al. (2002); Fally et al. (2003); Coheur et al. (2003). However, it is still challenging to measure those weak transitions located in the *near*-UV region with FTS instruments. Cavity ring-down spectroscopy (CRDS) offers both high sensitivity and high frequency precision, and the absorption coefficient can be determined directly without calibrating the optical path length. To the best of our knowledge,

the only high-resolution *near*-UV CRDS measurement of water was reported by (Dupré et al., 2005) in the 393 to 397 nm region. Meanwhile, a few laboratory measurements of cross-sections of water vapor in the *near*-UV region have been reported by different groups Pei et al. (2019); Wilson et al. (2016); Du et al. (2013), and several remote sensing studies Lampel et al. (2015, 2017) have also been carried out to detect *near*-UV water absorption, but large discrepancies exist in these results. In the earlier work by (Du et al., 2013), strong water vapor absorption cross-section values ranging from $2.94 \times 10^{-24}$ to $2.13 \times$

$10^{-25}$ cm$^2$molecule$^{-1}$ were observed with CRDS in the 290-350 nm region. Recent work from the same group Pei et al. (2019) presented quite different results, where water absorption cross-sections over this range were reported to be approximately $8.4 \times 10^{-25}$ to $1.6 \times 10^{-25}$ cm$^2$molecule$^{-1}$ by *near*-UV broadband CRDS. This work concluded that water vapor absorption significantly affects ozone retrievals and contributes $0.26 - 0.76$ Wm$^{-2}$ to the Earth's energy budget when incorporating the water cross-sections into a radiative transfer model Pei et al. (2019). The strong absorption of water vapor in the UV region

reported in these two studies Du et al. (2013); Pei et al. (2019) has been taken into account in ground-based measurements monitoring atmospheric trace gases of O$_3$, SO$_2$, and NO$_2$ Yin et al. (2021). It was found that the residual optical depth in the observed UV spectra is sensitive to the amount of atmospheric water vapor Yin et al. (2021). (Wilson et al., 2016) gave an upper limit for the water vapor absorption cross-sections of approximately $5 \times 10^{-26}$ cm$^2$molecule$^{-1}$ between 325 and 420 nm, which is different from the other two studies by (Du et al., 2013; Pei et al., 2019). (Lampel et al., 2015, 2017) also evaluated

the water vapor absorption in the *near*-UV region using multi-axis differential optical absorption spectroscopy (MAX-DOAS) and long-path (LP)-DOAS measurements, and a maximum cross-section value of $2.7 \times 10^{-27}$ cm$^2$molecule$^{-1}$ at 362.3 nm was demonstrated Lampel et al. (2017).

Generating a reliable water line list from the theoretical approach is also very challenging. A new water line list extending into the near-ultraviolet region $(42,000$ cm$^{-1})$ based on a semi-empirical potential energy surface (PES) and *ab initio*

dipole moment surface (DMS) was recently presented by (Conway et al., 2020). In particular, significant improvements in the *near*-UV region have been achieved for this line list, showing good agreement with recent atmospheric retrievals. The line list from (Conway et al., 2020) (supplemented with validated experimental results where possible) was used to update the HITRAN2020 database Gordon et al. (2022). The calculated line list is also added to the ExoMol database Tennyson et al. (2016) as well. Recently, by evaluating experimental rovibrational transitions a new database of the empirical energy levels called,

W2020 Furtenbacher et al. (2020) has been constructed. This database allowed some unobserved transitions to be predicted with experimental accuracy. For HITRAN2020, the W2020-derived line positions were used in place of *ab initio* results of (Conway et al., 2020) where possible. With that being said, very few transitions can be predicted in the region around 415 nm. This is because only one line at 24093.38 cm$^{-1}$ was assigned Tolchenov et al. (2005) in previous experimental studies in this





region and none of the measurements in other regions allow access to corresponding upper state energy levels. Therefore, the
majority of the line positions in HITRAN2020 in this region is still of *ab initio* origin.

Here, we report the high-resolution absorption spectroscopy of water vapor in the *near*-UV region measured by continuous-
wave cavity ring-down spectroscopy at room temperature. A sensitivity of $4 \times 10^{-10}$ cm$^{-1}$ was demonstrated, and the rovi-
brational transitions of water vapor were observed with intensities as weak as $10^{-28}$ cm molecule$^{-1}$. Assignments of these
observed lines are performed by comparing observed lines with the new *ab initio* water vapor line list from (Conway et al.,
2020). By comparing the observed spectra with simulated spectra from databases, the water vapor line lists in the *near*-UV
region from HITRAN2016 Gordon et al. (2017) and HITRAN2020 are evaluated. The recently updated HITRAN2020 line
list shows better agreement with the observed spectra in comparison with HITRAN2016. The water vapor absorption cross-
sections measured at room temperature around 415 nm are then convoluted to the same spectral resolution and compared with
references from previous observations and simulations.

## 2 Experimental

Water vapor absorption in the *near*-UV region was recorded by a continuous-wave cavity ring-down spectrometer. The config-
uration of our experimental setup is shown in Fig. 1 and briefly described as follows. An external cavity diode laser (ECDL,
Toptica DL Pro) was used as the light source. A small fraction of the laser beam is directly sent into a wavelength meter (High-
Finesse WS-7), with an absolute accuracy of 0.002 cm$^{-1}$. Another beam of about 10 mW is then coupled into a 0.74-m-long
ring-down cavity after an acousto-optical modulator (AOM). The ring-down cavity is composed of two high-reflectivity (HR)
mirrors ($R \approx 99.992\%$) with a piezoelectric actuator (PZT) which enables the control of the cavity length through a function
generator. The whole ring-down cavity is placed in a vacuum stainless chamber which is temperature-stabilized to 296.44 K
with a fluctuation of about 0.03 K. The ring-down event is initiated by a homemade trigger box that drives the AOM to switch
the laser beam, when the laser frequency matches the cavity mode and the transmitted laser power reaches a preset threshold.
The ring-down signal is detected by a photodiode and recorded through the data acquisition system. A nonlinear least-square
fitting algorithm is applied to derive the decay time $\tau_0$. Typically, about 200 decays are acquired to derive the decay time $\tau$
at a certain laser frequency. A single spectrum covering a range of 5 GHz can be recorded within 3 minutes. The resulting
noise-equivalent absorption coefficient is about $4 \times 10^{-10}$ cm$^{-1}$.

A deionized water sample with natural abundance was supplied to the cell through a needle valve. Since the exchange and
absorption between water vapor and the cavity walls are inevitable, the gas sample was stabilized for days until equilibrium was
reached before the measurement. The pressure inside the cavity was continuously monitored by a capacitance gauge (full-range
10 kPa, 0.5% accuracy) during the measurement. The initial pressure of the water sample inside the cavity was 1.12 kPa. And a
maximum fluctuation of the pressure during the measurement in 30 days was about 150 Pa, leading to a fractional uncertainty
of 2% of the sample pressure per day if we assumed a linear absorption of water vapor inside the cavity. Therefore, we corrected
the sample pressure of each recording based on the linear function of pressure on dates. An overview of the recorded spectra is
presented in Fig. 4(a). Water vapor absorption between 24062 and 24124 cm$^{-1}$ was recorded in this work.





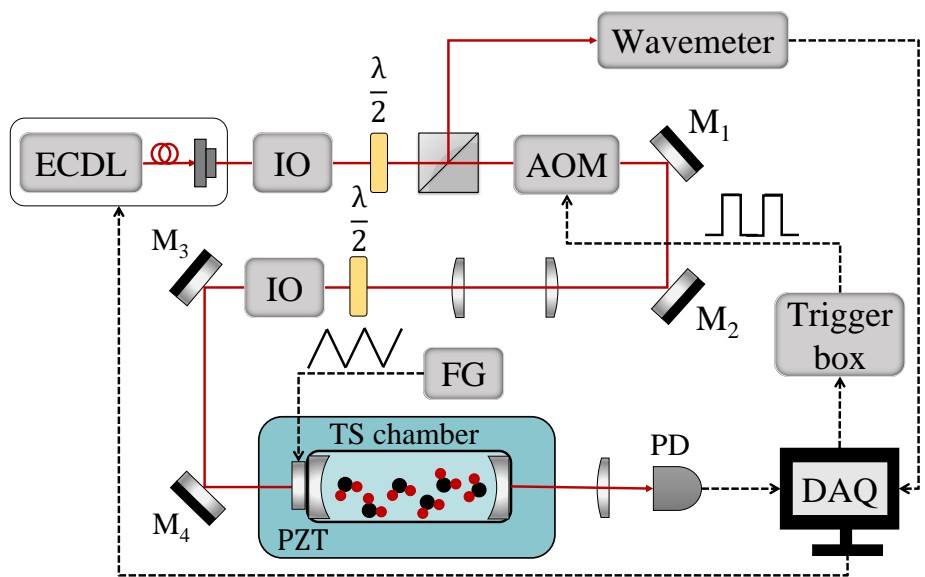

**Figure 1.** The experimental setup of the cavity ring-down spectroscopy. ECDL: external cavity diode laser, AOM: acousto-optical modulator, IO: optical isolator, PZT: piezoelectric actuator, DAQ: data acquisition system, FG: Function generator, TS chamber: Temperature Stabilized chamber.

## 3 Results and discussion

The spectral fitting procedure is outlined in Fig. 2. Two typical water absorption features are shown in Fig. 2 (a) and (b), including four transitions centered at 24082.3462 $cm^{-1}$, 24082.8808 $cm^{-1}$, 24093.3916 $cm^{-1}$ and 24093.6276 $cm^{-1}$. As shown in the upper panel of the figure, the original recorded spectra exhibited weak interference fringes on the baseline which may come from the optical back reflection from the cavity mirrors. A relatively stable free spectral range (FSR) of about 15 GHz was observed in our recorded original spectra which corresponds to an optical distance of 1 cm, while the amplitude of the interference fringes drifted over time. Therefore, a sine wave function was applied to fit the baseline drifting and then subtracted from the recorded spectra, as shown in the middle panel of Fig. 2 (c) and (d).

A simple Voigt profile was employed to fit the observed lines with a Gaussian line width fixed at the calculated Doppler broadening width. The bottom panels, Fig. 2 (e) and (f) show fitting residuals obtained with a Voigt profile, and a minimum detectable absorption coefficient of about $4 \times 10^{-10}$ $cm^{-1}$ which corresponds to a minimum detectivity of the cross-section around $1.5 \times 10^{-27}$ $cm^2$ molecule$^{-1}$ (see Fig. 5 (B)). Therefore, only transitions with line intensities larger than $2 \times 10^{-28}$ cm molecule$^{-1}$ were determined in this work, as summarized in Table. 1. In addition, the total cross-sections of wa-



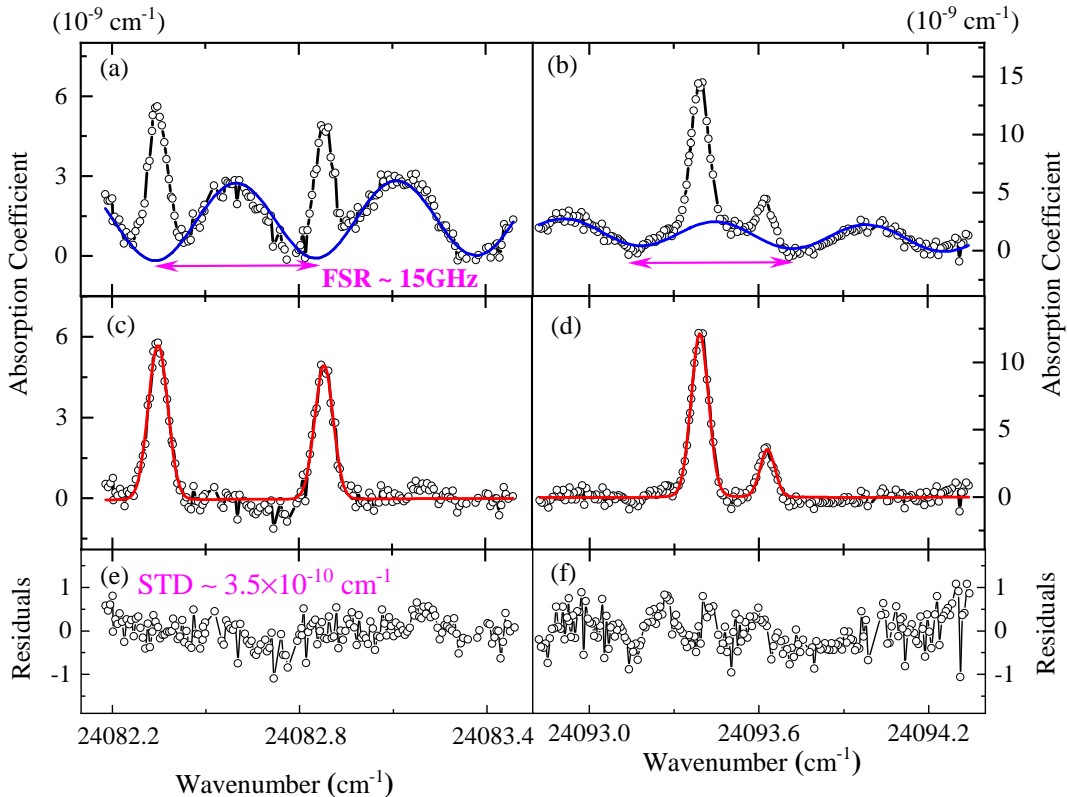

**Figure 2.** The recorded spectra of $H_2^{16}O$ around (a) 24082 cm$^{-1}$ and (b) 24093 cm$^{-1}$. Solid blue lines indicate the simulated fringes with a sine wave function. Spectra after subtraction of the simulated fringes are shown in (c) and (d). The absorption spectra of $H_2O$ obtained from a Voigt profile fitting are shown in solid red lines and their fitting residuals were plotted in (e) and (f). The standard deviation of the fitting residuals is about $4 \times 10^{-10}$ cm$^{-1}$.

ter vapor at 415 nm are presented in Fig. 5 (B), which has been convoluted to a spectral resolution of 5 cm$^{-1}$. In conclusion, we determined the cross-section values for water vapor around 415 nm varied between $1 \times 10^{-26}$ and $5 \times 10^{-26}$ cm$^2$ molecule$^{-1}$.

The calibration of the line positions given in Table. 1 is based on the wavemeter with absolute relative accuracy of 0.002 cm$^{-1}$, and the pressure in the sample cavity was around 1.12 kPa which contributed to the self-induced line shift of 0.0002 cm$^{-1}$ on average. As a result, the combined standard uncertainties around 0.002 cm$^{-1}$ are presented in Table. 1, which

are mostly limited by the wavelength calibration here. The values of the line intensities correspond to natural abundance of $H_2^{16}O$, and the relative uncertainties for the line intensities were included as well, seen in Table. 1. We have the relative uncertainty codes for 'A', 'B', and 'C', representing relative accuracy of $5-10\%$, $10-30\%$, and $>40\%$, respectively. The main uncertainties for the line intensity measurements came from the continuous exchange and absorption of the water molecules between the gas phase and the walls of the sample cell. The sample pressure in the cell was continuously monitored by a

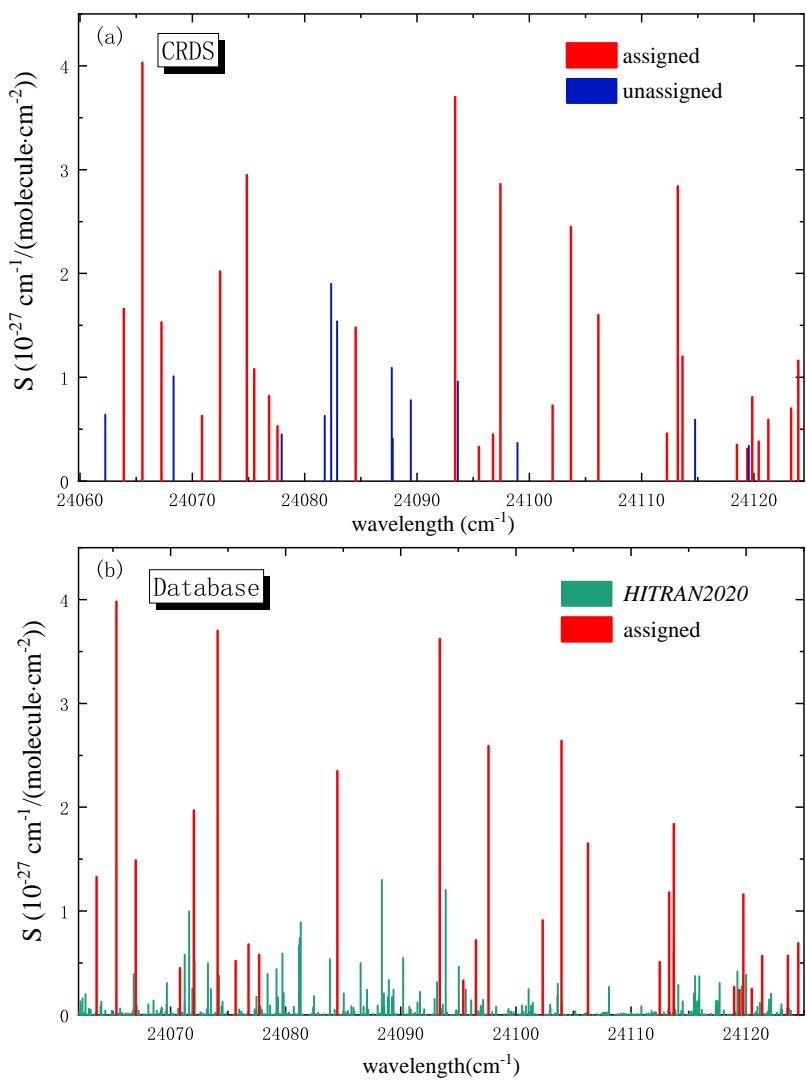

**Figure 3.** (a) An overview of the observed transitions in this work with red lines corresponding to the assigned transitions and blue lines corresponding to the unassigned transitions; (b) shows the assigned transitions in the red lines of (a) from (Gordon et al., 2022) (Also see Table 1 for more details).

capacitance gauge, which resulted in a maximum relative difference of about 2% every day during the measurements, as discussed before. In combination with the fitting uncertainties for the relatively strong absorption and weak absorption, and the temperature stability of our cavity, the intensity error codes were derived.





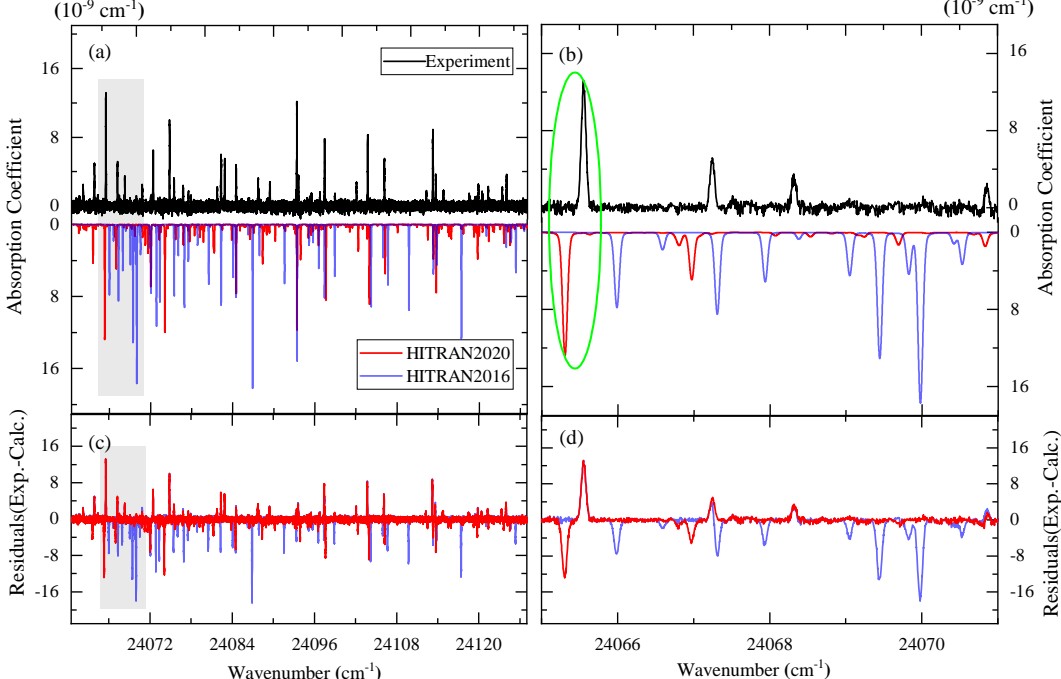

**Figure 4.** (a) Recorded and simulated spectra with different versions of the HITRAN database (including HITRAN2016 and HITRAN2020) between 24060 and 24130 cm$^{-1}$; (b) shows the zooms presented in the grey shadow of (a) ; (c) plots the residuals between the observed and calculated spectra; (d) shows the zooms presented in the grey shadow of (c). The black line corresponds to the experimental recordings, while the red and blue lines correspond to the simulations from HITRAN2020 and HITRAN2016 Gordon et al. (2017, 2022)

A line-by-line analysis of the observed spectra from 24,060 to 24130 cm$^{-1}$ was conducted through the direct comparison of experimental results with the *ab initio* line list from (Conway et al., 2020) which is also featured in the latest version of

the HITRAN database Gordon et al. (2022). Since there are fewer relative strong lines in this region compared to the low frequencies, only 40 transitions are observed in this work, as shown in Fig. 3(a).

(Conway et al., 2020) assigned part of their *ab initio* line list using the W2020 database of energy levels, but almost none of the transitions in this line list which are also within our region carry a full quantum assignment. These are defined as those transitions where upper and lower states carry both vibrational and rotational quantum labels. To address this, we attempt to

predict the upper states' assignment by considering the already known energy levels in W2020 together with the rotational quanta of states that are available in the line list of (Conway et al., 2020). The lower states are already assigned full vibrational and rotational quanta so we only focus on the upper states.

To begin, we group together all states in W2020 that have the same vibrational quanta and supplement these vibrational groups with the predicted/calculated vibrational quantum labels of the vibrational band origins (total angular momentum of





zero) given by (Császár et al., 2010). We then separate these vibrational groups into two more internal groups, defined by their rotational parity which can be ortho (triplet) or para (singlet). Finally, depending on the distribution, trajectory and quantity of the vibrational energy levels native to our ortho/para groups, we fit $J + K_a$ against energy (cm$^{-1}$) with either a two or three order polynomial, whichever gave the best fit. This was done as energy levels within a vibrational band and rotational parity typically vary smoothly. We know the upper state rotational quanta and parity from the line list of (Conway et al., 2020), hence

we can use these fits to determine an approximate value of the vibrational assignment.

    This procedure allows us to assign 27 new transitions, leaving 13 lines unassigned. The assigned lines are in good agreement with the predicted lines from the HITRAN database, also seen in Fig. 3(b). In addition, the standard deviation of differences between observed and calculated line positions is around 0.27 cm$^{-1}$, which is more than an order of magnitude larger than the experimental accuracy. No systematic shift between experimental and predicted line positions was observed, as can be seen in

Fig. 3 and Table 1. This discrepancy is likely due to the matched line positions of (Conway et al., 2020) being theoretical in origin. The observed line intensities differ from calculated values with a standard deviation of about 41%.

    A comparison of the simulated and measured spectra of water vapor is presented in Fig. 4. It indicates that there has been a significant improvement in this region for both line positions and intensities of HITRAN2020 compared with HITRAN2016. For instance, as shown in the zoom-in plot of Fig. 4 (b) and (c), there are many transitions in HITRAN2016 around 24070 cm$^{-1}$,

and the maximum line intensity is about $5 \times 10^{-27}$ cm molecule$^{-1}$. However, they were not observed in our measurements, and only a few very weak lines, being close to our noise level, are given in HITRAN2020. There are still considerable deviations between the experimental spectrum and the HITRAN2020 simulation. For instance, as indicated by the green circle on Fig. 4 (b), the line at 24065.5540(22) cm$^{-1}$ observed in this work corresponds to a line in HITRAN2020 with the position of 24065.304405 cm$^{-1}$ and a very close line intensity (see Table 1).

The discussion above gives us a general overview of the water vapor absorption in the *near*-UV region, and the comparison with the HITRAN database demonstrates that the line list for water vapor in the UV region has been substantially improved in the latest version. In order to make a further comparison of the cross-sections against other experimental results, we generated the line-by-line list from the CRDS measurements of our work and (Dupré et al., 2005), and compiled them with air-broadening coefficients ($\gamma_{air}$) from the HITRAN database. Water vapor cross-sections are then calculated through the

HITRAN Application Programming Interface (HAPI) Kochanov et al. (2016) convoluted with a spectral resolution of 5 cm$^{-1}$. Voigt profile was used in the calculation of the cross-sections. The result is depicted in Fig. 5, together with calculations from the HITRAN2020 Gordon et al. (2022) database in the range from 290 to 417 nm.

    The CRDS data around 396 nm reported by (Dupré et al., 2005) is also given in Fig. 5, as well as the cross-sections measured by (Pei et al., 2019), (Du et al., 2013) in the range of 290 - 350 nm. Both (Wilson et al., 2016) and (Lampel et al., 2017) have

placed upper limits of absorptions through different instrumental setups in this region as displayed in Fig. 5.

    The experimental cross-section data from this work and that from (Dupré et al., 2005) agree well with the calculated cross-sections based on HITRAN2020 data, as shown in Fig. 5(B) and (C). In addition, both CRDS results (this work and (Dupré et al., 2005)) fit in the upper limits given by (Wilson et al., 2016). However, there are large discrepancies in the range of 290 - 350 nm if we compare the HITRAN2020 data with experimental results from (Du et al., 2013) and (Pei et al., 2019). The



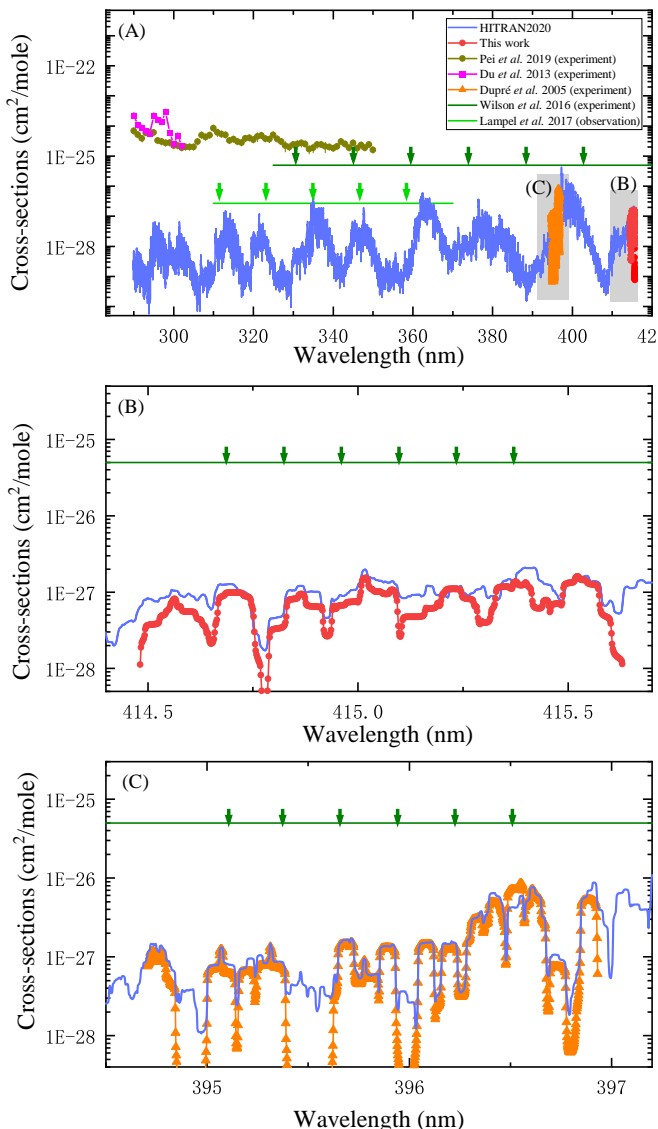

**Figure 5.** (A) An overview of cross-sections for water vapor obtained in this work and those reported in previous works, including the measurements by (Pei et al., 2019; Du et al., 2013; Dupré et al., 2005) and the upper limits by (Wilson et al., 2016; Lampel et al., 2017), as well as the calculations from (Gordon et al., 2022); (B) Zoomed plots for cross-sections determined in this work at 415 nm, and those simulated from HITRAN2020; (C) Zoomed plots for cross-sections simulated from (Dupré et al., 2005), and those from HITRAN2020. The simulated cross-sections plotted here are convoluted to a spectral resolution of 5 cm$^{-1}$ ($\approx 0.05$ nm) except that results from (Pei et al., 2019) are given in 1 nm step sizes and (Du et al., 2013) are given in 5 nm step sizes.





experimental results are more than two orders of magnitude higher than that given by the HITRAN database. And these results
are also at least 10 times higher than the other two limits reported by (Wilson et al., 2016) and (Lampel et al., 2017). It is
unlikely that the electronic state would affect the absorption in the 290-350 nm interval as described in (Conway et al., 2020),
since the nearest electronic state is far away from this region confirmed by different experiments Ranjan et al. (2020); Mota
et al. (2005); Chung et al. (2001). At this wavelength it is unlikely to observe electronic transitions without a considerable
population of highly excited vibration states which is not achievable at atmospheric temperatures. Hence we suggest that
further investigation of the water vapor absorption in the *near*-UV region is needed. And such measurement would greatly
benefit many other applications, such as the UV radiation measurements for monitoring $O_3$, $SO_2$, and $NO_2$ Yin et al. (2021),
and DOAS measurements of $O_4$, HONO, and OClO Lampel et al. (2017).

Table 1: The summary of line positions determined from the $H_2O$ spectrum recorded between 24062 and 24124 cm$^{-1}$ at
296.44($\pm$0.03) K. Uncertainties in brackets correspond to the unit of the last digit.

| Line position $\nu_0$(cm$^{-1}$) | | Intensity $S_0$ ($10^{-27} cm/mole$) | | assignments | | | |
|---|---|---|---|---|---|---|---|
| Expt. | Calc. | Expt. [1] | Calc. | $Vib'$ | $Vib''$ | $J'_{Ka'Kc'}$ | $J''_{Ka''Kc''}$ |
| 24062.2486(55) | | 0.64(B) | | NA[2] | | | |
| 24063.8859(30) | 24063.594790 | 1.66(A) | 1.33 | 2 2 4 | 0 0 0 | 3 3 1 | 2 2 0 |
| 24065.5540(22) | 24065.304405 | 4.03(A) | 3.98 | 2 2 4 | 0 0 0 | 3 3 0 | 2 2 1 |
| 24067.2467(30) | 24066.971030 | 1.53(A) | 1.49 | 2 0 5 | 0 0 0 | 2 0 2 | 1 0 1 |
| 24068.3227(38) | | 1.01(A) | | NA | | | |
| 24070.8573(55) | 24070.829724 | 0.63(B) | 0.45 | 2 0 5 | 0 0 0 | 3 1 3 | 2 1 2 |
| 24072.4556(26) | 24072.022238 | 2.02(A) | 1.97 | 7 1 0 | 0 0 0 | 6 3 4 | 5 2 3 |
| 24074.8387(23) | 24074.086363 | 2.95(A) | 3.70 | 2 2 4 | 0 0 0 | 4 3 2 | 3 2 1 |
| 24075.4931(38) | 24075.661374 | 1.08(A) | 0.52 | -2-2-2[3] | 0 0 0 | 3 3 0 | 2 2 1 |
| 24076.8213(64) | 24076.784250 | 0.82(B) | 0.68 | -2-2-2 | 0 0 0 | 7-3-3 | 6 1 6 |
| 24077.5833(49) | 24077.701694 | 0.53(B) | 0.58 | 3 0 4 | 0 0 0 | 4 1 3 | 3 2 2 |
| 24077.9427(74) | | 0.45(C) | | NA | | | |
| 24081.7717(55) | | 0.63(B) | | NA | | | |
| 24082.3462(28) | | 1.90(A) | | NA | | | |
| 24082.8808(28) | | 1.54(A) | | NA | | | |
| 24084.5446(30) | 24084.499892 | 1.48(A) | 2.35 | 0 9 3 | 0 0 0 | 4 2 2 | 3 0 3 |
| 24087.7444(40) | | 1.09(A) | | NA | | | |
| 24087.8350(74) | | 0.41(C) | | NA | | | |
| 24089.4547(49) | | 0.78(B) | | NA | | | |





| | | | | | | | |
|---|---|---|---|---|---|---|---|
| 24093.3916(22) | 24093.383000 | 3.70(A) | 3.62 | 7 1 0 | 0 0 0 | 4 4 1 | 3 3 0 |
| 24093.6276(38) | | 0.96(A) | | NA | | | |
| 24095.5171(92) | 24095.434499 | 0.33(C) | 0.33 | -2-2-2 | 0 0 0 | 8-3-3 | 7 3 4 |
| 24096.7601(63) | 24096.518347 | 0.45(B) | 0.72 | -2-2-2 | 0 0 0 | 5-3-3 | 4 1 4 |
| 24097.4297(26) | 24097.614315 | 2.86(A) | 2.59 | 7 1 0 | 0 0 0 | 5 3 2 | 4 2 3 |
| 24098.9361(74) | | 0.37(C) | | NA | | | |
| 24102.0763(45) | 24102.336665 | 0.73(B) | 0.91 | 2 2 4 | 0 0 0 | 5 4 2 | 4 3 1 |
| 24103.7063(24) | 24103.956924 | 2.45(A) | 2.64 | 2 2 4 | 0 0 0 | 5 4 1 | 4 3 2 |
| 24106.1421(28) | 24106.258560 | 1.60(A) | 1.65 | 7 1 0 | 0 0 0 | 6 4 3 | 5 3 2 |
| 24112.2552(63) | 24112.490751 | 0.46(B) | 0.51 | -2-2-2 | 0 0 0 | 6-1-1 | 5 3 3 |
| 24113.2126(24) | 24113.300066 | 2.84(A) | 1.18 | 1 2 5 | 0 0 0 | 4 3 1 | 3 1 2 |
| 24113.6354(40) | 24113.722387 | 1.20(A) | 1.84 | 2 2 4 | 0 0 0 | 5 5 0 | 4 4 1 |
| 24114.7532(86) | | 0.59(C) | | NA | | | |
| 24118.4875(92) | 24118.965600 | 0.35(C) | 0.27 | 5 3 1 | 0 0 0 | 0 0 0 | 1 0 1 |
| 24119.4089(92) | 24119.402890 | 0.31(C) | 0.24 | 1 2 5 | 0 0 0 | 3 3 1 | 2 1 2 |
| 24119.5389(92) | | 0.34(C) | | NA | | | |
| 24119.8424(49) | 24119.765909 | 0.81(B) | 1.16 | 5 3 1 | 0 0 0 | 6 5 2 | 5 4 1 |
| 24120.4196(92) | 24120.480591 | 0.38(C) | 0.25 | -2-2-2 | 0 0 0 | 8-4-4 | 7 4 3 |
| 24121.2748(63) | 24121.393473 | 0.59(B) | 0.57 | -2-2-2 | 0 0 0 | 7-3-3 | 6 3 4 |
| 24123.3144(55) | 24123.631663 | 0.70(B) | 0.57 | 7 1 0 | 0 0 0 | 7 5 2 | 6 4 3 |
| 24123.9541(36) | 24124.521673 | 1.16(A) | 0.69 | 7 1 0 | 0 0 0 | 6 6 1 | 5 5 0 |

[1] Codes of 'A', 'B' and 'C' correspond fractional uncertainties of 5-10%, 10-30% and > 40%, respectively;

[2] NA indicates unassigned transitions;

[3] Negative notations indicates unknown upper state.

## 4 Conclusion

In summary, we built a continuous-wave cavity ring-down spectroscopic setup and recorded the absorption spectra of water vapor in the *near*-UV region surrounding 415 nm. With a minimum detectable limit of $4 \times 10^{-10}$ cm$^{-1}$, 40 water vapor
transitions have been observed around 415 nm, with line intensities as low as $3 \times 10^{-28}$ cm/molecule. Assignments of the experimental data were conducted based on the *ab initio* calculation from (Conway et al., 2020). To the best of our knowledge, these absorption lines were never experimentally verified before, and 27 of them were assigned belonging to the (224), (205), (710), (304), (093), (125), and (531) vibrational bands of H$_2^{16}$O. The experimental line list is summarized in Table. 1 with



natural isotopic abundances. The accuracy of line positions determined in this work is about 60 MHz, and relative uncertainties of line intensities vary from 5-10%, 10-30% to $> 40\%$, depending on the line strengths. The recorded absorption spectra exhibit better agreement with calculations from HITRAN2020 than those from HITRAN2016. The results of this work will help improve the corresponding line positions in the HITRAN database, as well as a global network of water vapor energy levels in the next iteration of the W2020 database. Moreover, it will help empirical adjustments of future *ab initio* PES. Indeed, as explained in (Conway et al., 2020) points obtained from the first principles are fit together with the empirically-derived energy levels to an analytical function. That function is then used for calculating the line list. Since no energy levels were available in this region, it impacted the quality of the predicted line positions. Therefore, including the energy levels derived from our work will most certainly improve the predictive abilities of the semi-empirical PES. Adding air-broadening coefficients ($\gamma_{air}$) from the HITRAN database to the line parameters measured in this work, we produced the cross-sections of water vapor which is up to $1 \times 10^{-27}$ cm² molecule$^{-1}$ at a resolution of 0.05 nm. Comparisons are made against several experimental results of cross-sections in the *near*-UV region. The HITRAN2020 cross-sections are in good agreement with our results and those reported in the previous works of (Dupré et al., 2005) and (Wilson et al., 2016). However, they don't support recent suggestions of very strong absorption in the *near*-UV region from (Du et al., 2013) and (Pei et al., 2019).

*Data availability.* All data relevant to this study are available from the corresponding author upon reasonable request.

*Author contributions.* Q.-Y. Yang and H. Liang built the experimental setup and performed the experiments and spectral fitting procedure, Eamon K. Conway constructed and described the assignment. Iouli E. Gordon contributed the discussion of cross-sections with perspectives to the HITRAN database. Yan Tan and S.-M. Hu contributed to all the work in this paper and wrote most of it. All authors reviewed this paper and provided many corrections and suggestions.

*Competing interests.* The authors declare that they have no conflict of interest.

*Acknowledgements.* This work was jointly supported by the National Natural Science Foundation of China (41905018, 21903080, 21688102), and by the Chinese Academy of Sciences (XDC07010000).



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
