# Peer review of "Cavity ring-down spectroscopy of water vapor at deep blue region"

_Atmospheric Measurement Techniques, 2022_

## Author Response (AR1)

Response to reviewers:
We would like to thank both reviewers for their careful reading of the manuscript and their positive feedback and constructive criticism. It certainly helped us to improve the manuscript. We addressed the reviewers' comments in detail. Our responses are provided in blue italic font.

Reviewer#1:

The manuscript "Cavity ring-down spectroscopy of water vapor in the near-UV region" is well written and structured and adresses the important question of water vapour absorption in the near-UV region from about 33000-24000 cm-1 or 300-420nm. These absorption are typically not used to obtain water vapour concentrations in the atmosphere
or in experiments, but may overlay other absorbers and thus introduce systematic biases in measurements of various trace gases as mentioned in the manuscript. Water vapour is not the only absorber where work needs to be done for further advances in remote sensing applications, but often one of the strongest interferences in the near UV region. For other gases important progress was reported e.g. in Finkenzeller and Volkamer 2022 in the same spectral range.

Thanks for the positive comments. The detailed responses to the reviewer's comments are provided in the following part, which are presented in regular font whereas our responses are given in blue and when the text from the paper is quoted in blue italics.

Only a few comments follow:

- Figure 5: The line colour might be chosen differently to distinguish the upper limits by

Wilson et al and Lampel et al better. Lampel et al 2017 reported the upper limit only up to 350nm, this is wrong in the plot. The same publication also estimated the actual absorption cross-section around 363nm at a lower spectral resolution, which might also be included in the plot. The reported discreapancy there between observations and POKAZATEL was explained later in Conway et al 2020.

[Figure]

Thanks for pointing out the mistake in Figure5.
The color of those two upper limits by Wilson et al. and Lampel et al. have been changed to different colors now as shown revised Figure 5.
And we fixed the reported limits by Lampel et al.2017 in the region of 310-350nm at a 0.7 nm resolution. The absorption cross-section around 362.3nm was also added to the plot at a 0.5 nm resolution.

- Lampel et al 2015 estimated scaling factors for older HITRAN versions also for the spectral range around 400 and 415nm, but this might be difficult to include in the figure, and no dominating scaling factor for individual water vapour absorption lines listed in Table 1 between modelled and measured intensities can be seen.

We agree with the reviewer not to include the Lampel et al. 2015 to our plot, since the scalling factor were estimated from older HITRAN. And we already presented the direct comparison of the line-by-line spectral simulations of the most two recent HITRAN versions to our experimental results. Besides, the line parameters of both positions and intensities were listed in Table 1.

- Maybe also Conway et al 2020 or a recent line list could be included in the plot, as HITRAN based absorption cross sections were underestimating the actual absorptions especially in the UV due to a relatively large line-cutoff value.

[Figure]

As mentioned in our paper, the linelist for HITRAN2020 actually came from Conway et al.2020 (with some empirical corrections) with the intensity cutoff at 1E-30. And the intensity cutoff for Conway et al.2020 is 1E-32. So we have compared the simulated cross-sections in the range from 414-416nm at a resolution of 5cm-1(~0.05nm). The maximum difference is below 4E-29 cm2/mole which is much smaller than the precision of our experiment, so we decide not to include Conway et al.2020 in our plot.

- Henning Finkenzeller, Rainer Volkamer,O2–O2 CIA in the gas phase: Cross-section of weak bands, and continuum absorption between 297–500 nm,Journal of Quantitative Spectroscopy and Radiative Transfer,Volume 279,2022,108063,ISSN 0022-4073,https://doi.org/10.1016/j.jqsrt.

Thanks for the comment. The reference is included in the revised manuscript.

Reviewer#2:

Yang et al. describe a new cavity-ringdown instrument for high spectral resolution measurement of water vapour absorption around 415 nm. Despite the atmospheric abundance and radiative importance of water vapour, the experimental measurement of its absorption in the blue/near-UV spectral regions has proved surprisingly challenging. This work fills an important gap in our understanding of this key atmospheric species.

The spectral measurements reported here are a significant advance on earlier experimental studies which were limited by a lack of sensitivity. The work of Yang et al. provides indirect evidence that experimental studies showing strong near-UV absorption by water vapour need to be treated with caution. The results of the study are also relevant to recent theoretical work to inform the HITRAN database. The study provides a full quantum assignment to a number of the observed transitions. This work also points to the need to continue both theoretical and (especially) experimental investigations in the blue/near-UV absorption spectrum of water.

Thank you for acknowledging the importance of our work. This is most encouraging. The detailed responses to the reviewer's comments are provided in the following part, which are presented in regular font whereas our responses are given in blue and when the text from the paper is quoted in blue italics.

Comments:

- This paper describes spectra around 415 nm and labels this the "near-UV". 415 nm is firmly in the visible spectrum which extends from 400 to 700 nm according to most definitions. The generally accepted definition of near-UV extends up to 400 nm (e.g., UV-A: 315-400 nm). Their use of "near-UV" in referring to their measurements is therefore misleading. I recommend that the terminology throughout the manuscript, including in the title, be changed to reflect this distinction, for example, by replacing "near-UV" with "at deep blue wavelength" or something similarly appropriate elsewhere.

We have replaced "near-UV region" with "at deep blue" where we mentioned our experiment and in the title.

- Further experimental detail is needed. What is the frequency or period for acquisition between individual ringdown measurements? What is the wavelength interval and laser linewidth?

Thanks for the comment. The frequency between individual ringdown measurements is around 40Hz. The laser linewidth is around 100 kHz. The following sentences are now added to the manuscript.

*"An external cavity diode laser (ECDL,Toptica DL Pro) was used as the light source with a linewidth around 100 kHz. "*
*"Typically, about 200 decays for about 5 s are acquired to derive the decay time $\tau$ at a certain laser frequency. Therefore, the frequency for acquisition between individual ringdown measurements is about 40Hz."*

- Attribution of the interference fringes to the cavity mirrors seems reasonable but speculative. There are other system components that could give rise to the same effect, including the ½ wave plate and lenses. A more definitive identification of the source of the fringes should be provided.

Thanks for the comment. We attributed the interference fringes to the cavity mirrors since the amplitude of the fringes could be diminished if we adjust the angle of the cavity mirror and the FSR of 15 GHz also matched with the thickness of our mirrors. The following sentence is now added to the manuscript.

*"As shown in the upper panel of the figure, the original recorded spectra exhibited weak interference fringes on the baseline and the amplitude of the interference fringes drifted over time. We attributed these interference fringes to the optical back reflection from the cavity mirrors since the amplitude could be diminished if we adjust the angle of the cavity mirror, and a relatively stable free spectral range (FSR) of about 15 GHz was observed in our recorded original spectra which corresponds to an optical distance of 1 cm."*

▪ 4: "minimum detectable absorption coefficient of about 4*10−10 cm−1". This statement seems to confuse the standard deviation (precision) with a limit of detection (usually a specified factor of 2 or 3 greater than the measurement precision). Using the label "standard deviation of the residual", or the measurement precision, would be more correct.

Thanks for the suggestion. We have corrected the abstract and manuscript. The sentence is now updated as the following:

*"The bottom panels, Fig. 2 (e) and (f) show fitting residuals obtained with a Voigt profile, and a standard deviation of the residual of about $4 \times 10^{-10} cm^{-1}$ which also corresponds to minimum cross sections of $1.5 \times 10^{-27} cm^2 molecule^{-1}$."*

▪ The same reasoning should be applied to the "minimum detectivity of the cross section around 1.5*10−27 cm2 molecule−1" and the criterion for identifying a minimum line intensity should be stated explicitly.

Thanks for the suggestion. We have corrected the manuscript. The sentence is now updated as the following:

*"The bottom panels, Fig. 2 (e) and (f) show fitting residuals obtained with a Voigt profile, and a standard deviation of the residual of about $4 \times 10^{-10} cm^{-1}$ which also corresponds to minimum cross sections of $1.5 \times 10^{-27} cm^2 molecule^{-1}$."*

▪ 113: The authors attribute the main intensity uncertainty to exchange between the gas phase and container walls. Is this not just the uncertainty in pressure during the measurement? The main uncertainty seems to be the measurement precision relative to the signal (peak) size (alluded to in l.116). The dominant source of absorption uncertainty should be clarified.

Thanks for the suggestion. We have corrected the manuscript. The sentence is now updated as the following:

*"The main uncertainties for the line intensity measurements came from two parts, the first part is the continuous exchange and absorption of the water molecules between the gas phase and the walls of the sample cell. The sample pressure in the cell was continuously monitored by a capacitance gauge, which resulted in a maximum relative difference of about 2% every day during the measurements, as discussed before. And the temperature stability of our cavity is better than 0.05 %, which is negligible in the contribution of intensity uncertainties. In combination with the second part from fitting uncertainties for the relatively strong absorption and weak absorption, the intensity error codes were derived."*

- In this regard, it would be helpful to see example spectra of absorption lines in the B & C uncertainty category, not just type A absorption lines as shown in Fig. 2.

[Figure]

Thanks for the suggestion, we've updated Fig.2 which included not only type A absorption but also type B&C lines.

- 172: Although measurements of water vapour could affect the absorption spectra and retrieval of other trace gases, this and other studies of water vapour measurements imply that such potential interferences are unlikely — at least, at the current absorption sensitivities of atmospheric instrumentation.

Thanks for the suggestion, we've rephrased in the manuscript. The sentence is now updated as the following:

*"Our measurement, as well as other experimental results, have confirmed that water vapor absorption is very unlikely to affect the atmospheric retrievals for monitoring $O_3$, $SO_2$, and $NO_2$ (Yin et al., 2021), and DOAS measurements of $O_4$, HONO, and OClO (Lampel et al., 2017) at the current absorption sensitivities of atmospheric instrumentation."*

Minor textual issues:

- Citation style in the manuscript text should follow the standard AMT style and enclose citations in brackets. It is confusing to have references appear in the middle or end of sentences without any distinction between sentence text and citation text.

Thank you for pointing out the citation problem. We have fixed it in the revised manuscript.

- 90: It is unclear what is meant by "pressure on dates".

As we described in the corresponding paragraph, the pressure inside the cavity was continuously monitored by a capacitance gauge during our experiment about 30 days. We kept the sample cell as a closed cavity so that a fractional uncertainty of 2% of the sample pressure every day is determined if we assumed a linear absorption of water vapor inside the cavity. Therefore, we corrected the recordings to the pressure of the corresponding date.

- 174 (Table 1). Add "to": "correspond TO fractional uncertainties of 5-10%, 10-30% and > 40%,"

Fixed.